# GP CaKe: Effective brain connectivity with causal kernels

**Luca Ambrogioni**
Radboud University
l.ambrogioni@donders.ru.nl

**Max Hinne**
Radboud University
m.hinne@donders.ru.nl

**Marcel A. J. van Gerven**
Radboud University
m.vangerven@donders.ru.nl

**Eric Maris**
Radboud University
e.maris@donders.ru.nl

## Abstract

A fundamental goal in network neuroscience is to understand how activity in one brain region drives activity elsewhere, a process referred to as effective connectivity. Here we propose to model this causal interaction using integro-differential equations and causal kernels that allow for a rich analysis of effective connectivity. The approach combines the tractability and flexibility of autoregressive modeling with the biophysical interpretability of dynamic causal modeling. The causal kernels are learned nonparametrically using Gaussian process regression, yielding an efficient framework for causal inference. We construct a novel class of causal covariance functions that enforce the desired properties of the causal kernels, an approach which we call GP CaKe. By construction, the model and its hyperparameters have biophysical meaning and are therefore easily interpretable. We demonstrate the efficacy of GP CaKe on a number of simulations and give an example of a realistic application on magnetoencephalography (MEG) data.

## 1   Introduction

In recent years, substantial effort was dedicated to the study of the network properties of neural systems, ranging from individual neurons to macroscopic brain areas. It has become commonplace to describe the brain as a network that may be further understood by considering either its anatomical (static) scaffolding, the functional dynamics that reside on top of that or the causal influence that the network nodes exert on one another [1–3]. The latter is known as *effective connectivity* and has inspired a surge of data analysis methods that can be used to estimate the information flow between neural sources from their electrical or haemodynamic activity[2, 4]. In electrophysiology, the most popular connectivity methods are variations on the autoregressive (AR) framework  [5]. Specifically, Granger causality (GC) and related methods, such as partial directed coherence and directed transfer function, have been successfully applied to many kinds of neuroscientific data [6, 7]. These methods can be either parametric or non-parametric, but are not based on a specific biophysical model [8, 9]. Consequently, the connectivity estimates obtained from these methods are only statistical in nature and cannot be directly interpreted in terms of biophysical interactions [10]. This contrasts with the framework of dynamic causal modeling (DCM), which allows for Bayesian inference (using Bayes factors) with respect to biophysical models of interacting neuronal populations [11]. These models are usually formulated in terms of either deterministic or stochastic differential equations, in which the effective connectivity between neuronal populations depends on a series of scalar parameters that specify the strength of the interactions and the conduction delays [12]. DCMs are usually less flexible than AR models since they depend on an appropriate parametrization of the effective

connectivity kernel, which in turn depends on detailed prior biophysical knowledge or Bayesian model comparison.

In this paper, we introduce a new method that is aimed to bridge the gap between biophysically inspired models, such as DCM, and statistical models, such as AR, using the powerful tools of Bayesian nonparametrics [13]. We model the interacting neuronal populations with a system of stochastic integro-differential equations. In particular, the intrinsic dynamic of each population is modeled using a linear differential operator while the effective connectivity between populations is modeled using causal integral operators. The differential operators can account for a wide range of dynamic behaviors, such as stochastic relaxation and stochastic oscillations. While this class of models cannot account for non-linearities, it has the advantage of being analytically tractable. Using the framework of Gaussian process (GP) regression, we can obtain the posterior distribution of the effective connectivity kernel without specifying a predetermined parametric form. We call this new effective connectivity method *Gaussian process Causal Kernels* (GP CaKe). The GP CaKe method can be seen as a nonparametric extension of linear DCM for which the exact posterior distribution can be obtained in closed-form without resorting to variational approximations. In this way, the method combines the flexibility and statistical simplicity of AR modeling with the biophysical interpretability of a linear DCM.

The paper is structured as follows. In Section 2 we describe the model for the activity of neuronal populations and their driving interactions. In Section 3 we construct a Bayesian hierarchical model that allows us to learn the causal interaction functions. Next, in Subsection 3.2, we show that these causal kernels may be learned analytically using Gaussian process regression. Subsequently in Section 4, we validate GP CaKe using a number of simulations and demonstrate its usefulness on MEG data in Section 5. Finally, we discuss the wide array of possible extensions and applications of the model in Section 6.

## 2  Neuronal dynamics

We model the activity of a neuronal population $x_j(t)$ using the stochastic differential equation

$$\mathcal{D}_j x_j(t) = I_j(t) + w_j(t) \ , \tag{1}$$

where $I_j(t)$ is the total synaptic input coming from other neuronal populations and $w_j(t)$ is Gaussian white noise with mean 0 and variance $\sigma^2$. The differential operator $\mathcal{D}_j = \alpha_0 + \sum_{p=1}^{P} \alpha_p \frac{d^p}{dt^p}$ specifies the internal dynamic of the neuronal population. For example, oscillatory dynamic can be modeled using the damped harmonic operator $\mathcal{D}_j^H = \frac{d^2}{dt^2} + \beta \frac{d}{dt} + \omega_0^2$ , where $\omega_0$ is the (undamped) peak angular frequency and $\beta$ is the damping coefficient.

In Eq. 1, the term $I_j(t)$ accounts for the *effective connectivity* between neuronal populations. Assuming that the interactions are linear and stationary over time, the most general form for $I_j(t)$ is given by a sum of convolutions:

$$I_j(t) = \sum_{i=1}^{N} \big(c_{i \to j} \star x_i\big)(t) \ , \tag{2}$$

where the function $c_{i \to j}(t)$ is the causal kernel, modeling the effective connectivity from population $i$ to population $j$, and $\star$ indicates the convolution operator. The causal kernel $c_{i \to j}(t)$ gives a complete characterization of the linear effective connectivity between the two neuronal populations, accounting for the excitatory or inhibitory nature of the connection, the time delay, and the strength of the interaction. Importantly, in order to preserve the causality of the system, we assume that $c_{i \to j}(t)$ is identically equal to zero for negative lags ($t < 0$).

Inserting Eq. 2 into Eq. 1, we obtain the following system of stochastic integro-differential equations:

$$\mathcal{D}_j x_j(t) = \sum_{i=1}^{N} \big(c_{i \to j} \star x_i\big)(t) + w_j(t), \quad j = 1 \ldots N \ , \tag{3}$$

which fully characterizes the stochastic dynamic of a functional network consisting of $N$ neuronal populations.

## 3 The Bayesian model

We can frame the estimation of the effective connectivity between neuronal populations as a nonparametric Bayesian regression problem. In order to do this, we assign a GP prior distribution to the kernel functions $c_{i \to j}(t)$ for every presynaptic population $i$ and postsynaptic population $j$. A stochastic function $f(t)$ is said to follow a GP distribution when all its marginal distributions $p(f(t_1), \ldots, f(t_n))$ are distributed as a multivariate Gaussian [14]. Since these marginals are determined by their mean vector and covariance matrix, the GP is fully specified by a mean and a covariance function, respectively $m_f(t) = \langle f(t) \rangle$ and $\mathfrak{K}_f(t_1, t_2) = \langle (f(t_1) - m_f(t_1))(f(t_2) - m_f(t_2)) \rangle$. Using the results of the previous subsection we can summarize the problem of Bayesian nonparametric effective connectivity estimation in the following way:

$$
\begin{aligned}
c_{i \to j}(t) &\sim GP\left(0, \mathfrak{K}(t_1, t_2)\right) \\
w_j(t) &\sim N(0, \sigma^2) \\
\mathcal{D}_j x_j(t) &= \sum_{i=1}^{N} \left(c_{i \to j} \star x_i\right)(t) + w_j(t) \ ,
\end{aligned}
\tag{4}
$$

where expressions such as $f(t) \sim GP\big(m(t), \mathfrak{K}(t_1, t_2)\big)$ mean that the stochastic process $f(t)$ follows a GP distribution with mean function $m(t)$ and covariance function $\mathfrak{K}(t_1, t_2)$.

Our aim is to obtain the posterior distributions of the effective connectivity kernels given a set of samples from all the neuronal processes. As a consequence of the time shift invariance, the system of integro-differential equations becomes a system of decoupled linear algebraic equations in the frequency domain. It is therefore convenient to rewrite the regression problem in the frequency domain:

$$
\begin{aligned}
c_{i \to j}(\omega) &\sim CGP\big(0, \mathfrak{K}(\omega_1, \omega_2)\big) \\
w_j(\omega) &\sim CN(0, \sigma^2) \\
\mathcal{P}_j(\omega) x_j(\omega) &= \sum_{i=1}^{N} x_i(\omega) c_{i \to j}(\omega) + w_j(\omega) \ ,
\end{aligned}
\tag{5}
$$

where $\mathcal{P}_j(\omega) = \sum_{p=0}^{P} \alpha_p (-i\omega)^p$ is a complex-valued polynomial since the application of a differential operator in the time domain is equivalent to multiplication with a polynomial in the frequency domain. In the previous expression, $CN(\mu, \nu)$ denotes a circularly-symmetric complex normal distribution with mean $\mu$ and variance $\nu$, while $CGP(m(t), \mathfrak{K}(\omega))$ denotes a circularly-symmetric complex valued GP with mean function $m(\omega)$ and Hermitian covariance function $\mathfrak{K}(\omega_1, \omega_2)$ [15]. Importantly, the complex valued Hermitian covariance function $\mathfrak{K}(\omega_1, \omega_2)$ can be obtained from $\mathfrak{K}(t_1, t_2)$ by taking the Fourier transform of both its arguments:

$$
\mathfrak{K}(\omega_1, \omega_2) = \int_{-\infty}^{+\infty} \int_{-\infty}^{+\infty} e^{-i\omega_1 t_1 - i\omega_2 t_2} \mathfrak{K}(t_1, t_2) dt_1 dt_2 \ .
\tag{6}
$$

### 3.1 Causal covariance functions

In order to be applicable for causal inference, the prior covariance function $\mathfrak{K}(t_1, t_2)$ must reflect three basic assumptions about the connectivity kernel: I) temporal localization, II) causality and III) smoothness. Since we perform the GP analysis in the frequency domain, we will work with $\mathfrak{K}(\omega_1, \omega_2)$, i.e. the double Fourier transform of the covariance function.

First, the connectivity kernel should be localized in time, as the range of plausible delays in axonal communication between neuronal populations is bounded. In order to enforce this constraint, we need a covariance function $\mathfrak{K}(t_1, t_2)$ that vanishes when either $t_1$ or $t_2$ becomes much larger than a time constant $\vartheta$. In the frequency domain, this temporal localization can be implemented by inducing correlations between the Fourier coefficients of neighboring frequencies. In fact, local correlations in the time domain are associated with a Fourier transform that vanishes for high values of $\omega$. From Fourier duality, this implies that local correlations in the frequency domain are associated with a function that vanishes for high values of $t$. We model these spectral correlations using a squared exponential covariance function:

$$
\mathfrak{K}_{SE}(\omega_1, \omega_2) = e^{-\vartheta \frac{(\omega_2 - \omega_1)^2}{2} + it_s(\omega_2 - \omega_1)} = e^{-\vartheta \frac{\varsigma^2}{2} + it_s \varsigma} \ ,
\tag{7}
$$

where $\zeta = \omega_2 - \omega_1$. Since we expect the connectivity to be highest after a minimal conduction delay $t_s$, we introduced a time shift factor $it_s\zeta$ in the exponent that translates the peak of the variance from 0 to $t_s$, which follows from the Fourier shift theorem. As this covariance function depends solely on the difference between frequencies $\zeta$, it can be written (with a slight abuse of notation) as $\mathfrak{K}_{SE}(\zeta)$.

Second, we want the connectivity kernel to be causal, meaning that information cannot propagate back from the future. In order to enforce causality, we introduce a new family of covariance functions that vanish when the lag $t_2 - t_1$ is negative. In the frequency domain, a causal covariance function can be obtained by adding an imaginary part to Eq. 7 that is equal to its Hilbert transform $\mathcal{H}$ [16]. Causal covariance functions are the Fourier dual of quadrature covariance functions, which define GP distributions over the space of analytic functions, i.e. functions whose Fourier coefficients are zero for all negative frequencies [15]. The causal covariance function is given by the following formula:

$$\mathfrak{K}_C(\zeta) = \mathfrak{K}_{SE}(\zeta) + i\mathcal{H}\mathfrak{K}_{SE}(\zeta) \ . \tag{8}$$

Finally, as communication between neuronal populations is mediated by smooth biological processes such as synaptic release of neurotransmitters and dendritic propagation of potentials, we want the connectivity kernel to be a smooth function of the time lag. Smoothness in the time domain can be imposed by discounting high frequencies. Here, we use the following discounting function:

$$f(\omega_1, \omega_2) = e^{-\nu \frac{\omega_1^2 + \omega_2^2}{2}} \ . \tag{9}$$

This discounting function induces a process that is smooth (infinitely differentiable) and with time scale equal to $\nu$ [14]. Our final covariance function is given by

$$\mathfrak{K}(\omega_1, \omega_2) = f(\omega_1, \omega_2) \left( \mathfrak{K}_{SE}(\zeta) + i\mathcal{H}\mathfrak{K}_{SE}(\zeta) \right) \ . \tag{10}$$

Unfortunately, the temporal smoothing breaks the strict causality of the covariance function because it introduces leakage from the positive lags to the negative lags. Nevertheless, the covariance function closely approximates a causal covariance function when $\nu$ is not much bigger than $t_s$.

## 3.2 Gaussian process regression

In order to explain how to obtain the posterior distribution of the causal kernel, we need to review some basic results of nonparametric Bayesian regression and GP regression in particular. Nonparametric Bayesian statistics deals with inference problems where the prior distribution has infinitely many degrees of freedom [13]. We focus on the following nonparametric regression problem, where the aim is to reconstruct a series of real-valued functions from a finite number of noisy mixed observations:

$$y_t = \sum_i \gamma_i(t) f_i(t) + w_t \ , \tag{11}$$

where $y_t$ is the $t$-th entry of the data vector $y$, $f_i(t)$ is an unknown latent function and $w_t$ is a random variable that models the observation noise with diagonal covariance matrix $D$. The mixing functions $\gamma_i(t)$ are assumed to be known and determine how the latent functions generate the data. In nonparametric Bayesian regression, we specify prior probability distributions over the whole (infinitely dimensional) space of functions $f_i(t)$. Specifically, in the GP regression framework this distribution is chosen to be a zero-mean GP. In order to infer the value of the function $f(t)$ at an arbitrary set of target points $T^\times = \{t_1^\times, ..., t_m^\times\}$, we organize these values in the vector $\boldsymbol{f}$ with entries $f_l = f(t_l^\times)$. The posterior expected value of $\boldsymbol{f}$, that we will denote as $\boldsymbol{m}_{f_j|y}$, is given by

$$\boldsymbol{m}_{f_j|y} = K_{f_j}^\times \Gamma_j \left( \sum_i \Gamma_i K_{f_i} \Gamma_i + D \right)^{-1} \boldsymbol{y} \ , \tag{12}$$

where the covariance matrix $K_f$ is defined by the entries $[K_f]_{uv} = \mathfrak{K}_f(t_u, t_v)$ and the cross-covariance matrix $K_\psi^\times$ is defined by the entries $[K_f^\times]_{uv} = \mathfrak{K}_f(t_u^\times, t_v)$ [14]. The matrices $\Gamma_i$ are square and diagonal, with the entries $[\Gamma_i]_{uu}$ given by $\gamma_i(t_u)$.

It is easy to see that the problem defined by Eq. 5 has the exact same form as the generalized regression problem given by Eq. 11, with $\omega$ as dependent variable. In particular, the weight functions $\gamma_i(\omega)$ are given by $\frac{x_i(\omega)}{\mathcal{P}_j(\omega)}$ and the noise term $\frac{w_j(\omega)}{\mathcal{P}_j(\omega)}$ has variance $\frac{\sigma^2}{|\mathcal{P}_j(\omega)|^2}$. Therefore, the expectation of the posterior distributions $p(c_{i\to j}(\omega)|\{x_1(\omega_h)\}, \dots, \{x_N(\omega_h)\})$ can be obtained in closed from from Eq. 12.

# 4 Effective connectivity simulation study

We performed a simulation study to assess the performance of the GP CaKe approach in recovering the connectivity kernel from a network of simulated sources. The neuronal time series $x_j(t)$ are generated by discretizing a system of integro-differential equations, as expressed in Eq. 3. Time series data was then generated for each of the sources using the Ornstein-Uhlenbeck process dynamic, i.e.

$$\mathcal{D}^{(1)} = \frac{d}{dt} + \alpha \, , \tag{13}$$

where the positive parameter $\alpha$ is the relaxation coefficient of the process. The bigger $\alpha$ is, the faster the process reverts to its mean (i.e. zero) after a perturbation. The discretization of this dynamic is equivalent to a first order autoregressive process. As ground truth effective connectivity, we used functions of the form

$$c_{i \to j}(\tau) = a_{i \to j} \tau e^{-\frac{\tau}{s}} \, , \tag{14}$$

where $\tau$ is a (non-negative) time lag, $a_{i \to j}$ is the connectivity strength from $i$ to $j$ and $s$ is the connectivity time scale.

In order to recover the connectivity kernels $c_{i \to j}(t)$ we first need to estimate the differential operator $\mathcal{D}^{(1)}$. For simplicity, we estimated the parameters of the differential operator by maximizing the univariate marginal likelihood of each individual source. This procedure requires that the variance of the structured input from the other neuronal populations is smaller than the variance of the unstructured white noise input so that the estimation of the intrinsic dynamic is not too much affected by the coupling.

Since most commonly used effective connectivity measures (e.g. Granger causality, partial directed coherence, directed transfer function) are obtained from fitted vector autoregression (VAR) coefficients, we use VAR as a comparison method. Since the least-squares solution for the VAR coefficients is not regularized, we also compare with a ridge regularized VAR model, whose penalty term is learned using cross-validation on separately generated training data. This comparison is particularly natural since our connectivity kernel is the continuous-time equivalent of the lagged AR coefficients between two time series.

## 4.1 Recovery of the effective connectivity kernels

We explore the effects of different parameter values to demonstrate the intuitiveness of the kernel parameters. Whenever a parameter is not specifically adjusted, we use the following default values: noise level $\sigma = 0.05$, temporal smoothing $\nu = 0.15$ and temporal localization $\vartheta = \pi$. Furthermore, we set $t_s = 0.05$ throughout.

Figure 1 illustrates connectivity kernels recovered by GP CaKe. These kernels have a connection strength of $a_{i \to j} = 5.0$ if $i$ feeds into $j$ and $a_{i \to j} = 0$ otherwise. This applies to both the two node and the three node network. As these kernels show, our method recovers the desired shape as well as the magnitude of the effective connectivity for both connected and disconnected edges. At the same time, Fig. 1B demonstrates that the indirect pathway through two connections does not lead to a non-zero estimated kernel. Note furthermore that the kernels become non-zero after the zero-lag mark (indicated by the dashed lines), demonstrating that there is no significant anti-causal information leakage.

The effects of the different kernel parameter settings are shown in Fig. 2A, where again the method is estimating connectivity for a two node network with one active connection, with $a_{i \to j} = 5.0$. We show the mean squared error (MSE) as well as the correlation between the ground truth effective connectivity and the estimates obtained using our method. We do this for different values of the temporal smoothing, the noise level and the temporal localization parameters. Figure 2B shows the estimated kernels that correspond to these settings. As to be expected, underestimating the temporal smoothness results in increased variance due to the lack of regularization. On the other hand, overestimating the smoothness results in a highly biased estimate as well as anti-causal information leakage. Overestimating the noise level does not induce anti-causal information leakage but leads to substantial bias. Finally, overestimating the temporal localization leads to an underestimation of the duration of the causal influence.

Figure 3 shows a quantitative comparison between GP CaKe and the (regularized and unregularized) VAR model for the networks shown in Fig. 1A and Fig. 1B. The connection strength $a_{i \to j}$ was

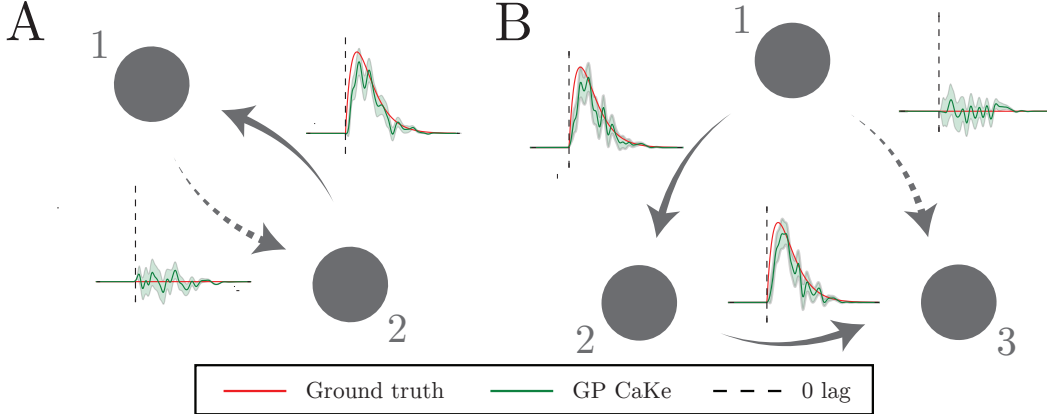

Figure 1: Example of estimated connectivity. **A.** The estimated connectivity kernels for two connections: one present ($2 \rightarrow 1$) and one absent ($1 \rightarrow 2$). **B.** A three-node network in which node 1 feeds into node 2 and node 2 feeds into node 3. The disconnected edge from 1 to 3 is correctly estimated, as the estimated kernel is approximately zero. For visual clarity, estimated connectivity kernels for other absent connections ($2 \rightarrow 1$, $3 \rightarrow 2$ and $3 \rightarrow 1$) are omitted in the second panel. The shaded areas indicate the 95% posterior density interval over 200 trials.

varied to study its effect on the kernel estimation. It is clear that GP CaKe greatly outperforms both VAR models and that ridge regularization is beneficial for the VAR approach. Note that, when the connection strength is low, the MSE is actually smallest for the fully disconnected model. Conversely, both GP CaKe and VAR always outperform the disconnected estimate with respect to the correlation measure.

## 5 Brain connectivity

In this section we investigate the effective connectivity structure of a network of cortical sources. In particular, we focus on sources characterized by alpha oscillations (8–12Hz), the dominant rhythm in MEG recordings. The participant was asked to watch one-minute long video clips selected from an American television series. During these blocks the participant was instructed to fixate on a cross in the center of the screen. At the onset of each block a visually presented message instructed the participant to pay attention to either the auditory or the visual stream. The experiment also included a so-called 'resting state' condition in which the participant was instructed to fixate on a cross in the center of a black screen. Brain activity was recorded using a 275 channels axial MEG system.

The GP CaKe method can be applied to a set of signals whose intrinsic dynamic can be characterized by stochastic differential equations. Raw MEG measurements can be seen as a mixture of dynamical signals, each characterized by a different intrinsic dynamic. Therefore, in order to apply the method on MEG data, we need to isolate a set of dynamic components. We extracted a series of unmixed neural sources by applying independent component analysis (ICA) on the sensor recordings. These components were chosen to have a clear dipolar pattern, the signature of a localized cortical source. These local sources have a dynamic that can be well approximated with a linear mixture of linear stochastic differential equations [17]. We used the recently introduced temporal GP decomposition in order to decompose the components' time series into a series of dynamic components [17]. In particular, for each ICA source we independently extracted the alpha oscillation component, which we modeled with a damped harmonic oscillator: $\mathcal{D}_j^H = \frac{d^2}{dt^2} + \beta \frac{d}{dt} + \omega_0^2$. Note that the temporal GP decomposition automatically estimates the parameters $\beta$ and $\omega_0$ through a non-linear least-squares procedure [17].

We computed the effective connectivity between the sources that corresponded to occipital, parietal and left- and right auditory cortices (see Fig. 4A) using GP CaKe with the following parameter settings: temporal smoothing $\nu = 0.01$, temporal shift $t_s = 0.004$, temporal localization $\vartheta = 8\pi$ and noise level $\sigma = 0.05$. To estimate the causal structure of the network, we performed a $z$-test on the maximum values of the kernels for each of the three conditions. The results were corrected

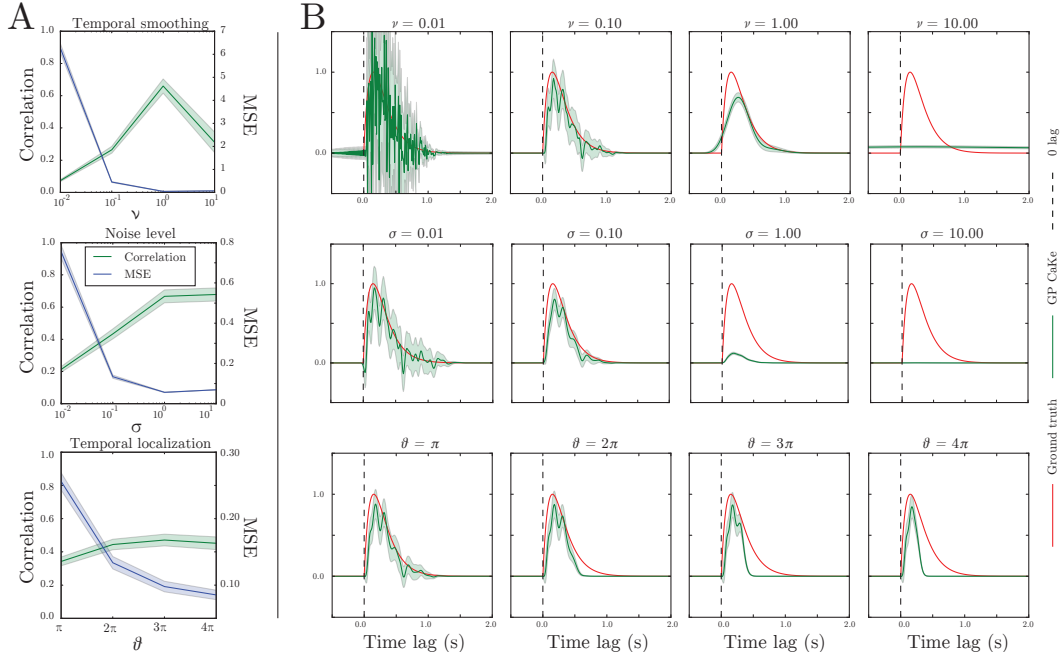

Figure 2: The effect of the the temporal localization, smoothness and noise level parameters on a present connection. **A.** The correlation and mean squared error between the ground truth connectivity kernel and the estimation by GP CaKe. **B.** The shapes of the estimated kernels as determined by the indicated parameter. Default values for the parameters that remain fixed are $\sigma = 0.05$, $\nu = 0.15$ and $\vartheta = \pi$. The dashed line indicates the zero-lag moment at which point the causal effect deviates from zero. The shaded areas indicate the 95% posterior density interval over 200 trials.

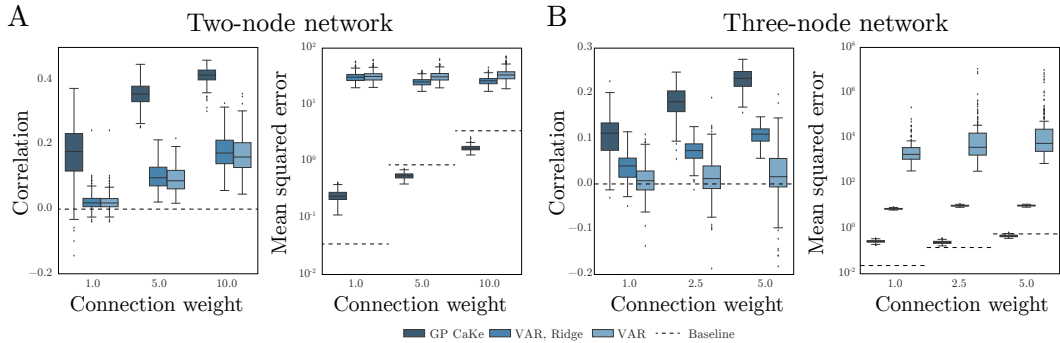

Figure 3: The performance of the recovery of the effective connectivity kernels in terms of the correlation and mean squared error between the actual and the recovered kernel. Left column: results for the two node graph shown in Fig. 1A. Right column: results for the three node graph shown in Fig. 1B. The dashed line indicates the baseline that estimates all node pairs as disconnected.

for multiple comparisons using FDR correction with $\alpha = 0.05$. The resulting structure is shown in Fig. 4A, with the corresponding causal kernels in Fig. 4B. The three conditions are clearly distinguishable from their estimated connectivity structure. For example, during the auditory attention condition, alpha band causal influence from parietal to occipital cortex is suppressed relative to the other conditions. Furthermore, a number of connections (i.e. right to left auditory cortex, as well as both auditory cortices to occipital cortex) are only present during the resting state.

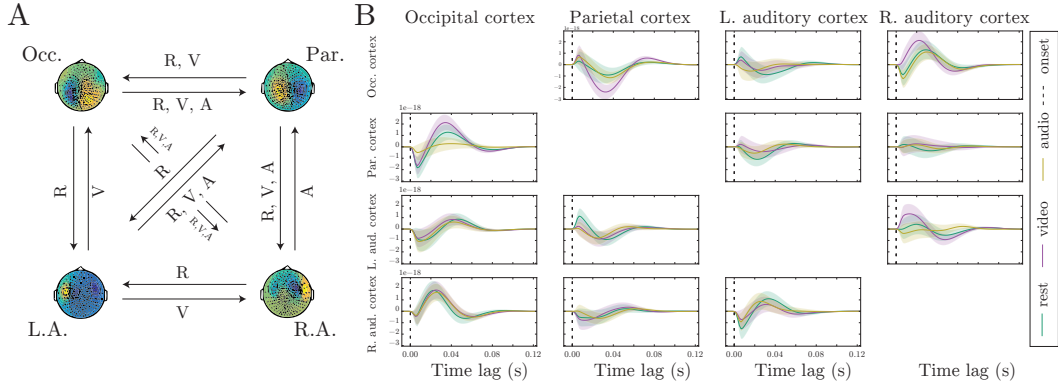

Figure 4: Effective connectivity using MEG for three conditions: I. resting state (R), II. attention to video stream (V) and III. attention to audio stream (A). Shown are the connections between occipital cortex, parietal cortex and left and right auditory cortices. **A.** The binary network for each of the three conditions. **B.** The kernels for each of the connections. Note that the magnitude of the kernels depends on the noise level $\sigma$, and as the true strength is unknown, this is in arbitrary units.

## 6    Discussion

We introduced a new effective connectivity method based on GP regression and integro-differential dynamical systems, referred to as GP CaKe. GP CaKe can be seen as a nonparametric extension of DCM [11] where the posterior distribution over the effective connectivity kernel can be obtained in closed form. In order to regularize the estimation, we introduced a new family of causal covariance functions that encode three basic assumptions about the effective connectivity kernel: (1) temporal localization, (2) causality, and (3) temporal smoothness. The resulting estimated kernels reflect the time-modulated causal influence that one region exerts on another. Using simulations, we showed that GP CaKe produces effective connectivity estimates that are orders of magnitude more accurate than those obtained using (regularized) multivariate autoregression. Furthermore, using MEG data, we showed that GP CaKe is able to uncover interesting patterns of effective connectivity between different brain regions, modulated by cognitive state.

The strategy for selecting the hyperparameters of the GP CaKe model depends on the specific study. If they are hand-chosen they should be set in a conservative manner. For example, the temporal localization should be longer than the highest biologically meaningful conduction delay. Analogously, the smoothing parameter should be smaller than the time scale of the system of interest. In ideal cases, such as for the analysis of the subthreshold postsynaptic response of the cellular membrane, these values can be reasonably obtained from biophysical models. When prior knowledge is not available, several off-the-shelf Bayesian hyperparameter selection or marginalization techniques can be applied to GP CaKe directly since both the marginal likelihood and its gradient are available in closed-form. In this paper, instead of proposing a particular hyper-parameter selection technique, we decided to focus our exposition on the interpretability of the hyperparameters. In fact, biophysical interpretability can help neuroscientists construct informed hyperprior distributions.

Despite its high performance, the current version of the GP CaKe method has some limitations. First, the method can only be used on signals whose intrinsic dynamics are well approximated by linear stochastic differential equations. Real-world neural recordings are often a mixture of several independent dynamic components. In this case the signal needs to be preprocessed using a dynamic decomposition technique [17]. The second limitation is that the intrinsic dynamics are currently estimated from the univariate signals. This procedure can lead to biases when the neuronal populations are strongly coupled. Therefore, future developments should focus on the integration of dynamic decomposition with connectivity estimation within an overarching Bayesian model.

The model can be extended in several directions. First, the causal structure of the neural dynamical system can be constrained using structural information in a hierarchical Bayesian model. Here, structural connectivity may be provided as an a priori constraint, for example derived from diffusion-weighted MRI [18], or learned from the functional data simultaneously [19]. This allows the model to automatically remove connections that do not reflect a causal interaction, thereby regularizing

the estimation. Alternatively, the anatomical constraints on causal interactions may be integrated into a spatiotemporal model of the brain cortex by using partial integro-differential neural field equations [20] and spatiotemporal causal kernels. In addition, the nonparametric modeling of the causal kernel can be integrated into a more complex and biophysically realistic model where the differential equations are not assumed to be linear [12] or where the observed time series data are filtered through a haemodynamic [21] or calcium impulse response function [22].

Finally, while our model explicitly refers to neuronal populations, we note that the applicability of the GP CaKe framework is in no way limited to neuroscience and may also be relevant for fields such as econometrics and computational biology.

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
