[Reviews · NeurIPS 2017]

Reviewer 1



This paper addresses the problem of understanding brain connectivity (i.e. how activity in one region from the brain drives activity in other regions) from brain activity observations. More generally, perhaps, the paper attempts to uncover causal structure and uses neuroscience insights to specifically apply the model to brain connectivity. The model can be seen as an extension of (linear) dynamic causal models (DCMs) and assumes that the observations are a linear combination of latent activities, which have a GP prior, plus Gaussian noise (Eq 11). Overall the paper is readable but more clarity in the details of how the posterior over the influence from i->j is actually computed (paragraph just below Eq 12). I write this review as a machine learning researcher (i.e. non-expert in neuroscience) so I am not entirely sure about the significance of the contribution in terms of neuroscience. For the machine learning perspective, the paper does not really provide a significant contribution as it applies standard GP regression. Indeed, it seems that the claimed novelty arises from the specific kernel used in order to incorporate localization, causality, and smoothness. However, none of these components seem actually new as they use, respectively, (i) the squared exponential (SE) covariance function, (iii) a previously published kernel, and (iii) some additional smoothing also previously known. With this regards, it is unclear why this additionally smoothing is necessary as the SE kernel is infinitely differentiable. The method is evaluated on a toy problem and on a real dataset. There are a couple of major deficiencies in the approach from the machine learning (ML) perspective. Firstly, the mixing functions are assumed to be known. This is very surprising as these functions will obviously have an important effect on the predictions. Second, for the only real dataset used, most (if not all) the hyperparameters are fixed, which also goes against standard ML practice. These hyperparameters are very important, as shown in the paper for the toy dataset, and ultimately affect the connectivities discovered. There also seems to be a significant amount of pre-processing to actually fit the problem to the method, for example, as explained in section 5, running ICA. It is unclear how generalizable such an approach would be in practice.

Reviewer 2



This nicely written paper presents a Bayesian nonparametric model to infer effective connectivity from neuroimaging data represented as integro-differential equations. The main contribution of the paper is the introduction of causal kernels for Gaussian processes that (at least approximately) satisfy certain properties so that they can be used to claim effective (and not simply functional) connectivity. The model specification is conjugate so that the posterior over interaction kernels is given in closed form. The method is demonstrated on both synthetic data and on real MEG data. Overall, I think this is a very good paper. However, there are a few parts of the paper where more discussion or more thorough descriptions could help make it more accessible and useful to a broader audience. - First, a more thorough explanation as to why adding the imaginary Hilbert transform to Eq. (8) enforces causality would be welcome. - The authors mention time-shift invariance to justify Eq. (5), but it's not clear from what's written where this comes from. - Why is the infinite differentiability of the functions a reasonable assumption? Why not OU or Matern or something that has some amount of smoothness? - In the experiments, was the smoothness parameter learned? The inferred kernels look pretty rough, would a fully Bayesian treatment where the smoothness hyperparmeter is learned help with this overfitting? - How were the hyperparameters chosen for the MEG experiment? - The authors point out that a limitation of the model is that linear differential operators need to be used. What breaks down if we break this assumption? Is linearity just needed for analytic tractability? Or is there a statistical reason? - How does the method scale with the number of signals?

Reviewer 3



This paper proposed a statistical model for brain connectivity analysis, based on stochastic differential equations and Gaussian processes. The assumptions and constraints are well described and justified, and the model is carefully designed based on conditions. Experimental results are provided on synthetic and real data, which justified the modeling power and interpretability of the model. 1. Is it true that the convolution causal kernel a sufficient characterization for connectivity? The triggering signal from i to j can simply be a spiking event/signal, while the intensity or accumulated signal/amplitude is irrelevant. 2. As much of the model formulation and analysis is done in frequency domain, it is helpful and more straightforward for the readers if some experiment results can also be provided in frequency domain. 3. Have you tried the model on signals including multiple frequency bands, like alpha and gamma bands? As different sources often appear in different frequency bands, it would be useful to model the connectivity involving those. 4. How are the hyper parameters are chosen in real-world data? For example, the temporal smoothing, localization and noise levels. 5. In a quite relevant work [1], different brain states are modeled using GPs with spectral kernels, like resting states and different sleeping states. Spatial-temporal correlation among brain regions and states are analyzed, which should be helpful to the authors. [1] K. Ulrich, D. E. Carlson, W. Lian, J. S. Borg, K. Dzirasa, and L. Carin. Analysis of brain states from multi-region LFP time-series. NIPS, 2014.